# A Portable Device for Methane Measurement Using a Low-Cost Semiconductor Sensor: Development, Calibration and Environmental Applications

**DOI:** 10.3390/s21227456

**Published:** 2021-11-10

**Authors:** Leonardo Furst, Manuel Feliciano, Laercio Frare, Getúlio Igrejas

**Affiliations:** 1Centro de Investigação de Montanha (CIMO), ESA, Instituto Politécnico de Bragança, 5300-253 Bragança, Portugal; leonardofurst@gmail.com; 2Department of Biological and Environmental Sciences, Federal Technological University of Paraná, Medianeira 85884-000, Brazil; laercio@utfpr.edu.br; 3Research Centre in Digitalization and Intelligent Robotics (CEDRI), Polytechnic Institute of Bragança, 5300-253 Bragança, Portugal; igrejas@ipb.pt

**Keywords:** gas sensor, semiconductor, greenhouse gases, methane, gas monitoring

## Abstract

Methane is a major greenhouse gas and a precursor of tropospheric ozone, and most of its sources are linked to anthropogenic activities. The sources of methane are well known and its monitoring generally involves the use of expensive gas analyzers with high operating costs. Many studies have investigated the use of low-cost gas sensors as an alternative for measuring methane concentrations; however, it is still an area that needs further development to ensure reliable measurements. In this work a low-cost platform for measuring methane within a low concentration range was developed and used in two distinct environments to continuously assess and improve its performance. The methane sensor was the Figaro TGS2600, a metal oxide semiconductor (MOS) based on tin dioxide (SnO_2_). In a first stage, the monitoring platform was applied in a small ruminant barn after undergoing a multi-point calibration. In a second stage, the system was used in a wastewater treatment plant together with a multi-gas analyzer (Gasera One Pulse). The calibration of low-cost sensor was based on the relation of the readings of the two devices. Temperature and relative humidity were also measured to perform corrections to minimize the effects of these variables on the sensor signal and an active ventilation system was used to improve the performance of the sensor. The system proved to be able to measure low methane concentrations following reliable spatial and temporal patterns in both places. A very similar behavior between both measuring systems was also well noticeable at WWTP. In general, the low-cost system presented good performance under several environmental conditions, showing itself to be a good alternative, at least as a screening monitoring system.

## 1. Introduction

Over the last two hundred years, the rapid increase in Earth’s average surface temperature has been caused mainly by human activities, due to the rise in atmospheric concentrations of the so-called greenhouse gases, of which carbon dioxide (CO_2_), water vapor (H_2_O), methane (CH_4_) and nitrous oxide (N_2_O) are the most impactful [1]. The excess of these gases increases the infrared radiation retained in the atmosphere, amplifying the greenhouse effect and raising the Earth’s surface average temperature [1,2].

Between the years 1750 and 2011, the average global concentration of methane increased by a factor of 2.5, raising from 722 ± 25 ppb to 1803 ± 4 ppb, mainly due to anthropogenic activities related to cattle raising; the expansion of rice crops; gaseous emissions from landfills; and the extraction, production, and use of fossil fuels. It is also estimated that 30% of the total emissions–anthropogenic and natural–are related to geological losses and losses in the natural gas production chain [3].

In the Fifth Assessment Report [3], the Intergovernmental Panel on Climate Change (IPCC) reported that methane has an estimated lifetime of about 12 years in the atmosphere, with a Global Warming Potential (GWP) 86 times higher than carbon dioxide over 20 years, and 34 times higher over 100 years. Besides the GWP, methane is one of the precursors of tropospheric ozone, which in turn is a strong oxidant detrimental to human health and vegetation [4,5]. Thus, tools for monitoring methane are essential for emissions control, air quality assessment and health protection.

Several technologies have been used to monitor air pollutants. The traditional methodologies involve optical, colorimetric and chromatographic techniques [6]. In environments with low methane concentrations, the most used methods are gas chromatography with flame ionization (GC-FID) [7] and cavity ring-down spectrometry (CRDS) [8]. These methods require dedicated instrumentation such as gas analyzers and carrier gases, which, due to their complexity, involves high acquisition, operation and maintenance costs. Despite the costs, these instruments have high accuracy and precision and are usually installed in strategic locations or in mobile labs for determination of regional trends and emissions [9]. However, for local or spot measurements, the use of gas analyzers becomes costly and requires a specialized technician to perform the task. Therefore, there is a growing interest involving low-cost sensors that can monitor gases with high reliability and incorporated into portable systems [10] or in monitoring networks [11].

One of the types of low-cost sensors used for monitoring methane is the sensors based on semiconductors [12]. These sensors determine the methane concentration by resistance variations proportional to the variations in the gas concentration. These sensors have become the target of several studies due to its low cost, flexibility, easy handling, and simple adaptation to various prototyping platforms. However, there are still several challenges in the application of low-cost sensors compared to gas analyzers, which raise several doubts about its performance and data quality. The main limitations of low-cost sensors are technological factors related to lifetime, selectivity and accuracy [13,14].

Another problem faced when measuring methane using semiconductor sensors is the presence of interfering gases. Interfering gases are considered pollutants to which the sensor has sensitivity, the most common being carbon monoxide, propane and ethanol [15]. A study performed with several SnO_2_-based sensors [15] showed that sensors of this type were able to react to different concentrations of methane, hexane, hydrogen, carbon monoxide, ammonia, hydrogen sulphide and ethanol. Thus, the use of SnO_2_-based sensors requires a previous knowledge of the application site to determine the possible interfering gases.

In this work, the potential of a low-cost sensor, the TGS2600 from Figaro, was studied when incorporated into a portable system for methane monitoring at two distinct environments—in barns and in indoor atmospheres of a wastewater treatment plant. Although several papers present the use of low-cost gas sensors [15,16,17] in different situations, commercially available low-cost sensors, mainly used in the detection of low concentrations of air pollutants, need more improvements and specific strategies for their reliable application. Thus, this work also covers the technical aspects related to the application of the semiconductor sensor TGS2600 for the detection of low concentrations of methane, discussing its limitations and the necessary details for a better field application, providing relevant information for the use of this type of sensor. Two methodologies of calibration are discussed, one involving a multi-point concentration and the other an inter-comparison with a more reliable and robust commercial system. The sensitivity of the sensor to other gases and signal deviations due to temperature and relative humidity was also studied.

## 2. Materials and Methods

### 2.1. System Architecture

The system was developed with the integration of several sensors on the Arduino Mega platform, including one air temperature and relative humidity sensor (Sensirion SHT31-D), one pressure sensor (Bosch BMP180), one GPRS module (SIMCOM SIM808), one SD card module, one 40 mm fan for air transport and one methane sensor (Figaro TGS2600). Figure 1 shows the system architecture with the interactions between each element.

The Arduino Mega is a low-cost, open source platform that allows users to access the system design and use it in a wide range of applications [18]. Compared to other prototyping platforms of the same manufacturer, the Arduino Mega has the advantage of having a greater number of input/output ports–54 digital, 16 analog–providing connection with a larger number of peripherals. Moreover, it is a module based on the ATmega2560 microcontroller, with 256 KB of flash memory, making it possible to compile larger codes [19].

Concerning methane concentrations, it was opted to use the TGS2600, a MOS (Metal Oxide Semiconductor) sensor from Figaro based on tin dioxide (SnO_2_) as a sensing material, as the technical specifications of the sensor indicated its ability to respond to methane concentrations within the desired range. Methane is present in the atmosphere at low concentrations, with a global background concentration around 1.8 ppm [3], and studies accomplished in Washington [20] and Boston [21] identified values of methane in urban areas varying between 2.5 and 90 ppm. The same sensor was used by Eugster and Kling [22] for measurements of methane concentrations in Toolik Lake, Alaska, where the sensor was able to identify concentrations of atmospheric methane below 2 ppm; and by Riddick et al. [16] to measure methane concentrations at one onshore gas terminal, identifying methane concentrations from 1.82 to 5.40 ppm.

The TGS2600 is a semiconductor sensor, capable of detecting methane, carbon monoxide, iso-butane, ethanol and hydrogen. The sensor has a high dependence on temperature and relative humidity (Figure 2), and corrections are required to obtain reliable results [23]. For this reason, to measure air temperature and relative humidity, an air temperature and relative humidity sensor SHT31-D Sensirion sensor was also implemented. This sensor can perform temperature measurements in the range of −40 °C to 125 °C with typical accuracy of ±0.3 °C and resolution of 0.015 °C and relative humidity (RH) between 0 and 100% with typical accuracy of ±2.0% RH and resolution of 0.01%. Despite the temperature and relative humidity range aforementioned, the manufacturer recommends measurements within the ranges 5 to 60 ° C for air temperature and 20 to 80% for relative humidity from to obtain results with more precision [24].

In addition, additional components were incorporated into the system: a BMP180 barometric sensor; a SIM808 GPS/GPRS/GSM module for acquisition of geographical position and communication between the system and a remote server using the mobile network; a microSD module to data storage; and a fan as an active air circulation system. Initially, tests were conducted without any air conduction system, so the sensors operated passively. However, this approach showed low efficiency, even with a source of methane very close to the sensors during the laboratory tests. As such, a 40 mm fan was incorporated in the system to force the ambient air to circulate on the sensors as showed in Figure 3. The use of active systems as vacuum pumps has been reported in works using the same sensor series [17,22], but a fan was chosen for this study due its low consumption and simplicity of installation. Eugster and Kling [22] assumed that although the methane sensor operates passively, forcing the air circulating on it could improve the signal resolution, especially when sampling at low wind speeds.

The configuration of the TGS2600 sensor used in this study was the basic reference circuit presented in the sensor technical documentation [23,25], applying the same value for the load resistance (R_L_) used in the work of Eugster and Kling [22]—a precision resistance of 5 kΩ/1%. Figure 4 shows the basic circuit of the sensor and its structure.

Sensor terminals 1 and 4 are responsible for powering the heating system, referenced as a heating resistor (R_H_), in charge of heating the sensor to the operating temperature of the device, promoting the reactions with the target gas. The heater material is RuO_2_. The sensor detection system is powered through terminals 2 and 3 and operates similarly to an adjustable resistor (R_S_), since its value varies according to the concentration of the target gas. The detection system is composed by two electrodes connected to the tin dioxide layer (sensing material). The load resistance (R_L_) is used as a reference to calculate the system voltage drop [23].

The detection principle of TGS2600 is based on variations in the resistance value caused by chemical reactions—adsorption, oxidation, diffusion—that occur between the electrode surface and the ambient air. As the temperature increases, adsorption of the atmospheric oxygen on the crystalline surface of the electrode occurs in the form of O^2−^ ions, forming a kind of barrier that prevents the movement of electrons, increasing the resistance of the system. When the sensor is exposed to the target gas, an oxidation reaction occurs involving the gas and the O^2−^ ions, releasing the oxide ions which was bound to the surface, increasing the electron flow and decreasing the resistance of the sensor [26,27]. For this reason, when switched on, the TGS2600 sensor needs a stabilization time to reach the ideal temperature for the reactions to occur, where an increase in the sensor resistance is observed.

### 2.2. Signal Conditioning and Sensor Calibration

The TGS2600 is an analog sensor in which the concentration of methane is measured through a resistance ratio R_s_ (sensor resistance)/R_0_ (sensor resistance in methane fresh air), so that, some steps are needed to convert the signal into methane concentration (ppm). The output voltage signal provided by the sensor was collected through the analog port and converted to decimal values from 0 to 1023. The possible decimal values for the conversion are limited by the resolution of the internal 10-bit Arduino Analog-to-Digital Converter (ADC). Since the output voltage of the sensor is related with the methane concentration, the result of the ADC, which is a digital representation of the input voltage, is also related to the methane concentration. Thus, it is possible to establish a relation between the digital value and the measured methane concentrations. This can be done by using the reference voltage (5 V) and the digital resolution of the ADC, obtaining the equivalent voltage for each digital value.

When the sensor output voltage was determined, it was necessary to convert the voltage value to the resistance unit (Ohm), applying Equation (1), which is available in the TGS2600 technical document [25]. The equation establishes a relation between the supply voltage (Vc), the output voltage (Vout) and the load resistance (R_L_) to calculate the resistive value of the sensor (Rs), which corresponds to a variation of its resistance as a function of the change in methane concentration:(1)$$R_{s}=\frac{V_{C}{-V}_{\mathrm{out}}}{V_{\mathrm{out}}}-R_{L}$$

Knowing the value of Rs, the next step was the determination of the resistance in fresh air (R_0_) since the concentration of methane is calculated through a ratio of Rs/R_0_. The manufacturer specifies a maximum ratio of Rs/R_0_ = 1 where R_0_ represents the resistance measured in fresh air under controlled conditions of temperature and relative humidity of 20° C and 65%. The resistance observed in these conditions was adopted as R_0_. Thus, in the construction of the calibration curve and during the tests performed, an Rs/R_0_ ratio greater than 1 was obtained, indicating only the displacement of the sensor response line, since all the results are in relation to R_0_. This procedure and its effect on the Rs/R_0_ ratio were similar to those reported by Eugster and Kling [22].

The conversion of Rs/R_0_ into a methane concentration was firstly based on a multi-point calibration and at a later stage on an intercomparison test between the low-cost system and a multi-gas analyzer (Gasera One Pulse). The multi-point calibration involved the MCZ-MK5 multi-gas calibration system, which allowed the generation of airflows with different methane concentrations—0, 6.2, 12.2, 18.6, 29.8, 49.6, 69.4, 74.4, 111.6, 198.4, 297.6, 396.8, 437.6, 496 and 744 ppm—from a methane standard contained in a pressurized cylinder. The data were then processed to obtain the variation of the sensor resistance as a function of the methane concentrations, i.e., the calibration curve.

Taking into account the main results from the first tests carried out in the small ruminant barn, it was decided to calibrate the sensor by comparing its readings with those provided by the commercial system Gasera, allowing also a more rigorous evaluation of the low-cost sensor. The intercomparison test has involved parallel measurements of indoor methane concentrations in a Wastewater Treatment Plant facilities using both the low-cost system and Gasera One Pulse, which analyzes the infrared spectrum of the sample gases using a photoacoustic sensor based on the cantilever-enhanced optical microphone.

Finally, considering the high dependence of the sensor on temperature and relative humidity of the air, a correction model was applied. The model was constructed based on the sensor dependency characteristics presented by the manufacturer in its reference document [25]. A fourth-degree interpolation was applied for each dataset, resulting in three known polynomials, representing the relative air humidity curves of 40, 65 and 85%. Subsequently, linear interpolation was used between the polynomials, allowing the correction for multiple values of relative humidity of the air. Figure 5 shows some of the curves constructed through the adjustment process.

### 2.3. System Development

In the preliminary phase, the system was assembled using breadboards (Figure 6a,b), on which it was possible to add components and electronic terminals without welding, facilitating the modification, adaptation and removal of the components according to the characteristics of the desired application. In this phase a laboratory DC power supply was used to power the system.

After the calibration of the TGS2600 sensor and the preliminary tests, the system was modified to become a more compact and portable system (Figure 6b), dispensing the use of a fixed source of energy, using a nickel/cadmium battery of 12 V and 5 Ah. For this, the system components were organized to reduce the space and the voltage was regulated through a voltage regulator set to a 5 V output. The fan responsible for air circulation to the sensors was replaced by a smaller diameter fan, changing from 100 mm to a 40 mm. The system was then stored inside a plastic container for easy transportation and handling during field samplings.

After defining all the components used in the system, a printed circuit board (PCB) was developed. The layout and design of the board were developed in the Eagle software. The PCB was designed to use a 7.4 V lithium polymer rechargeable battery to provide power to four voltage regulators LM1085, responsible for adjusting the voltage from 7.4 V to 5 V, 4 V, 3.3 V and 3 V, required to feed the various sensors used in the application, and the battery charge control was performed using the LT3652 power tracking battery charger. The PCB was designed to be stackable, coupled directly onto the Arduino MEGA. At the top of the board was also installed the GPRS module, which was used to collect geographic coordinates and send data every 15 s to the IoT (Internet of Things) platform ThingSpeak for data storage and visualization. The system was inserted inside a wooden casing with dimensions of 140 × 145 × 70 mm^3^, facilitating the transportation and use. In addition, an LCD monitor was installed for real time data visualization. The Figure 7 shows the final system model.

### 2.4. Environmental Applications of the Low-Cost Monitoring System

For a more rigorous and complete evaluation of the reliability and robustness of the system, two types of field experiences were carried out in environments where relatively low and highly variable methane concentrations might be expected.

The first experiments were conducted in a small ruminant barn (sheep and goats) of 120 m^2^ of the High School of Agriculture of the Polytechnic Institute of Bragança, to analyze the response of the sensor to the presence of the gas. The site was defined by the extensive published literature, which proves the emissions from the digestive processes of ruminant animals.

Several sampling trials were performed in the barn. The first was conducted on 20 April 2017, without confined animals, lasting 35 min and having a sampling rate of 2 s. The second experiment was carried out on 21 April, with the presence of animals, lasting 60 min long and having a sampling rate of 2 s. A third test was conducted at the site on May 15, totaling 210 min in which the concentration of methane in the barn without the presence of animals and with subsequent entry of the animals was verified, to be able to analyze the signal change for different situations. The tests were conducted in alternating cycles of exposure to fresh air–with the system at six meters from the barn entrance–and methane, to observe the variation of the sensor signal for the different atmospheres. In this environmental application, the Rs/R_0_ ratio was converted into ppm of methane through the calibration curve obtained from the calibration methodology based on the use of a methane standard pressurized cylinder described in Section 2.2.

The second experience was carried out in October 2020 under similar conditions to that used for the intercomparison test for system calibration—at the belt filter press of a wastewater treatment plant and indoor workplaces to check the sensor response in different environments and methane concentrations. In this environmental application, the Rs/R_0_ ratio was converted into ppm of methane through the calibration curve provided by the intercomparison test with the Gasera One Pulse gas analyzer. This calibration was carried out to update the sensor calibration and because we believe it might provide a better calibration curve for mixed atmospheres with potential chemical interfering agents.

## 3. Results and Discussion

### 3.1. Calibration and Intercomparison Tests

Figure 8 presents the calibration curve based on the multi-point calibration, described in Section 2.2. This curve stablishes the relationship between R_S_/R_0_ values and the methane concentrations generated by the calibration system. Intervals corresponding to the sensor recovery after exposed to a high methane concentration were excluded.

In this figure it is observed that the R_S_/R_0_ ratio was higher than 1, indicating the displacement of the sensor response. Thus, the same R_0_ of the calibration curve should be used for the application, so that all R_S_ values are related to the same resistance in methane free fresh air.

Figure 9 shows the calibration curve based on the intercomparison between the TGS2600 and the Gasera One Pulse gas analyzer carried out at the Wastewater Treatment Plant, in different indoor locations. The outliers were identified and removed using standardized residuals. The new curve was used for measurements at the Wastewater Treatment Plant.

Both calibrations showed that the sensor can present different resistances for the same methane concentrations, showing low repeatability of the sensor. Thus, the first curve has the tendency to overestimate low methane concentrations due to its calibration having more points above 100 ppm, while the second curve tends to underestimate high values due to the points being below 100 ppm. It is also understood that, due to the first calibration being performed in the absence of other gases, the sensor response may differ from what is observed in its real application.

The calibration and intercomparison tests also demonstrate that a better curve fit can be reached according to the desired end application, based on the possible range of methane concentration for the study site. There is also the possibility of creating a sensor signal conditioning based on multiple curves, each intended for a range of methane. However, the implementation of this option is not linear, because signals of the same intensity are observed for high and low concentrations.

### 3.2. Application of Low-Cost Sensor at a Small Ruminant Barn

Figure 10 shows the methane concentration in the small ruminant barn without correction and after applying the correction for air temperature and relative humidity. The correction effect is well visible in the figure, mainly in the regions of the highest values of methane and relative humidity, showing that its application ensures a greater reliability of the results. The first level observed in the graphs corresponds to the start of the system, a period necessary to promote the heating of the sensor and, therefore, the stabilization of the signal. It is observed in the figure that the sensor signals increased significantly when kept inside the barn. The decrease of the signal in fresh air occurs smoothly, a tendency that corresponds to the regeneration time of the sensor after exposure to the target gas, which during the test phase ranged from 3 to 8 min. In the first sampling cycle, the methane average concentration measured in the barn was 33.7 ppm and during the second cycle, an average concentration of 234.3 ppm was observed.

The difference in values between the first and second exposures cycles is believed to be related to the instability of the atmosphere inside the barn, since methane production/release could not be uniform on the surface and the facilities have openings that promote non-homogeneous ventilation. Subsequent figures are presented only with the corrected values.

The second test is shown in the Figure 11. The high concentration difference between the two tests is attributed to the presence of the sheep during the second test, which, when moving, favors the release of the methane gas of the floor covering. In addition, they contribute to the elevation of methane concentration by releasing the gases produced during the digestion process. Another phenomenon observed in the second test was that even in the fresh air high concentrations of methane were detected. These high values reflect the lower dilution rate of atmosphere in the vicinity of the barn compared with dilution conditions prevailing in the first test, probably because of the low occurrence of wind gusts in the measurement day. From the first to the third exposure cycle of the sensor to methane gas, the sensor identified average concentrations of 5769.9 ppm, 6149.9 ppm and 7124.4 ppm, respectively, while the outdoor methane concentration observed after each exposure was 1805.0 ppm, 1938.8 ppm and 2047.8 ppm. The measured values were above the range covered by the calibration curve, so that the obtained values may deviate from the true values, as the sensor response is not linear.

The third test, conducted on 15 May (Figure 12), shows the evolution of methane concentration in the barn before and after confinement of the animals. The concentration of methane rises rapidly with the entrance of the sheep into the enclosure, going from a level of 50 ppm to values that exceed 1000 ppm. Some discrepant points were also observed, which deviate from the tendency of the curve and exceeded 3000 ppm. One hypothesis addressed for these variations is that the manufacturer does not specify the sensor behavior for relative humidity values below 35% in the equipment’s technical specifications [25], indicating greater variability of the sensor response in that range. This hypothesis was mentioned by Eugster and Kling [22], who verified that the behavior of the sensor does not obey the decreasing pattern of R_s_/R_0_ when elevating the temperature value to a constant range of relative humidity. The mean concentration observed during the absence of the animals was 35.3 ppm and, in the presence of the animals, a concentration of 591.3 ppm was obtained.

On 15 May 2017, the test was also identified lower concentrations of methane in relation to 20–21 April; this occurred because of the change of the floor covering, which consists of straw bedding. Manure deposition occurs in the straw bedding, influencing the production of methane due to the aging and accumulation of the manure.

Analyzing the results using the span calibration, it was found that the concentrations measured by the sensor were well above that observed in other works focused on measuring methane in barns. A study [28] measuring methane emissions of cows in a free stall barn during a milking procedure found peak concentrations ranging from 400 to 900 ppm. Another study [29] measuring gaseous emissions in a hoop grower-finisher swine barn identified maximum methane concentrations of approximately 15 ppm. Other research [30] with more similarity to this work, carried out in a sheep shed, found methane values ranging from 35.2 to 336 ppm. This was one of the reasons to realize comparative tests in order to obtain a more suitable calibration curve for atmospheres with potential interfering gases.

### 3.3. Application of the Low-Cost System in Indoor Atmospheres of a Wastewater Treatment Plant

Figure 13 shows the results obtained with both monitoring systems installed in the wastewater sludge dewatering room of a wastewater treatment plant. The peaks above 100 ppm are not shown since these values were only detected by the sensor. Nevertheless, the sensor presented good behavior, although with a mean relative error of about 39%. Regarding ammonia, a few moments of elevation of its concentration were observed, the most remarkable after 39 h, with a maximum value of 2.84 ppm. The average ammonia concentration was 1.07 ± 0.43 ppm. Pearson correlation coefficient test between TGS2600 methane concentration and ammonia showed that the variables were not significantly correlated at a confidence level of 0.05, with a *p*-value of 0.18, a possible indication that ammonia has no effect on the sensor signal.

Figure 14 shows the measurements performed indoors, in the office and laboratory of the wastewater treatment plant. In the three tests it was observed that the concentration measured by the sensor was on average 2 ppm below the reference concentrations. The same behavior was identified in the three tests. The average relative errors were about 40%, 35% and 25%, from left to right. This divergence could be lowered by applying an offset to approximate their values as the curves follow the same behavior. For ammonia, the average values were 1.02 ± 0.3, 0.91 ± 0.29 and 1.08 ± 0.45 ppm, from left to right. Using the Pearson correlation coefficient test, a significant correlation value was found at 0.05 of 0.18, 0.19 and 0.28, respectively. Although significant, the correlation value was low and when compared with the test of Figure 13 that did not show significant correlation, it is an indication that for higher methane concentrations, ammonia does not have a substantial effect on the sensor signal and for lower concentrations ammonia has none to low interference on the TGS2600 signal. This observation is in line with the technical specifications of the sensor, which indicated no cross-sensitivity to ammonia.

Despite the observed results, the use of the sensor requires that some care is necessary for the acquisition of reliable data, the correction of the signal as a function of the temperature and relative humidity of the air being essential, as shown in the technical documents of the sensor and verified in other works. For this reason, the use of the sensor SHT31 for measurement of these variables was necessary for posterior signal correction. In addition, the application of the sensor should be avoided near sources of carbon monoxide to prevent interference in the signal and requires the application of a stable voltage close to 5 V to prevent variations of power supply and consequently deviation in the sensor response.

The sensor accuracy could be improved using an analog converter with higher resolution. Considering that the Arduino has a 10-bit ADC, it means that it is capable of 1024 analog values, so when using the TGS2600, it will have 1024 possible responses from the sensor for its operating range. However, when using a 16-bit ADC, the system would be able to detect 65,536 analog values for the same operating range, and the system would be able to identify smaller variations in methane concentration. Another option to improve the accuracy at low concentrations of methane is the use of amplifiers as seen in studies using semiconductor gas sensors [31,32]. The use of amplifiers would allow the sensor to be more sensitive to lower concentrations, increasing signal intensity.

Calibration is also a fundamental part of the sensor application. One study [16] found that at low methane concentrations the signal from different TGS2600 sensors tend to vary for the same concentration, so that the application of a generalized calibration becomes unfeasible for obtaining reliable data. Moreover, the same authors suggest a regular calibration of the sensor since they found a drift of the sensor signal over time. Compared with studies that identified better results for low methane concentrations [16,22], it is understood that it would be necessary to apply the sensor for longer periods of time at methane concentrations not exceeding 10 ppm, in order to obtain a more specific curve for low concentrations.

## 4. Conclusions

This study aimed at developing a portable and low-cost system for methane measurement based on the Arduino platform using the TGS2600 sensor. To assess its performance, the sensor was used for measuring methane gas in a barn of small ruminants of the High School of Agriculture of the Polytechnic Institute of Bragança and in different indoor environments of a Wastewater Treatment Plant. The sensor has high dependence on temperature and humidity, which implies the measurement of relative humidity and temperature for signal corrections. Moreover, the cross sensitivity of the sensor limits its use to locations that do not have sources of carbon monoxide, iso-butane, ethanol, and hydrogen.

The measurements performed in the barn showed that the sensor has a good response to methane gas, showing an increase in signal when inside the barn and demonstrating that the presence of animals creates an atmosphere richer in methane. At the wastewater treatment plant, the sensor also showed a good response to concentrations below 50 ppm. Regarding ammonia, considered as an interfering gas in works using semiconductor sensors, the sensor showed low sensitivity to the observed ammonia concentrations.

The study also found the need for regular calibrations, because of the deviations in the sensor signal over time due to the sensor degradation, and the use of a ventilation system for air movement will improve the response of TGS2600 to methane. Regardless the restrictions in the use of the sensor, the system is promising as a screening system for preliminary or less stringent measurements.

The use of a signal amplifier with the TGS2600 sensor could improve the resolution of the sensor for lower concentrations. However, the amplification of the sensor signal implies in the increase of the signal noise; therefore, it would be necessary to perform tests with different amplifier gains to identify which one exhibits the best response with less noise.

## Figures and Tables

**Figure 1 sensors-21-07456-f001:**
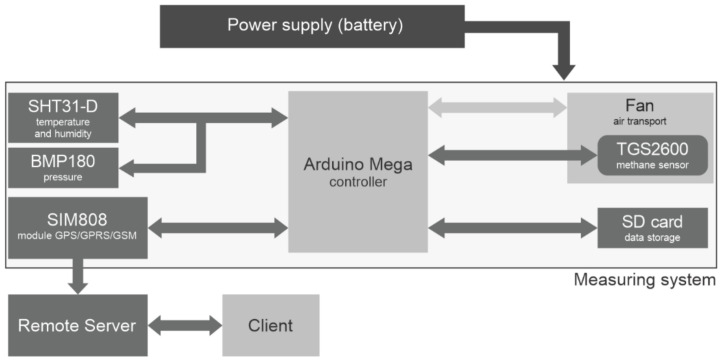
System architecture. The arrows represent the communication among the different elements of the system.

**Figure 2 sensors-21-07456-f002:**
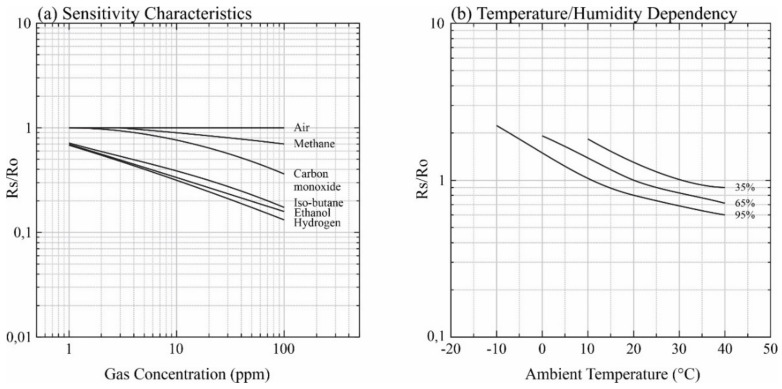
(**a**) Response of the Figaro TGS 2600 sensor to different gases; (**b**) Dependency of the Figaro TGS 2600 on temperature and relative humidity, where R_s_ is the resistance of the sensor to different concentrations of the target gas and R_0_ the resistance of the sensor in fresh air at 20 °C and 65% RH. adapted from [25].

**Figure 3 sensors-21-07456-f003:**
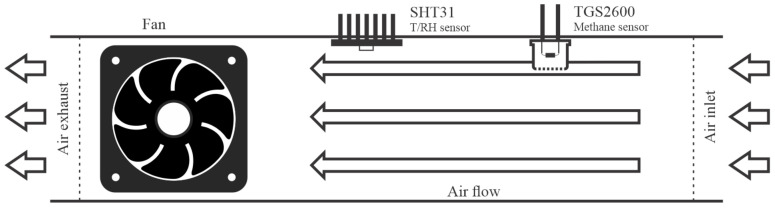
Active air circulation used in the system to improve signal resolution. The temperature and relative humidity sensor were installed next to the methane sensor.

**Figure 4 sensors-21-07456-f004:**
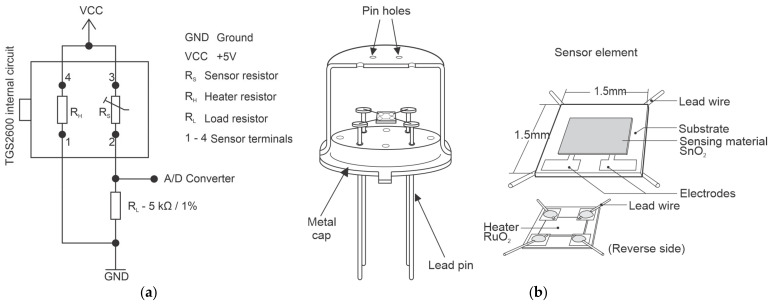
Figaro TGS 2600 basic sensor: (**a**) conditioning circuit; (**b**) internal and external structure. Adapted from [23].

**Figure 5 sensors-21-07456-f005:**
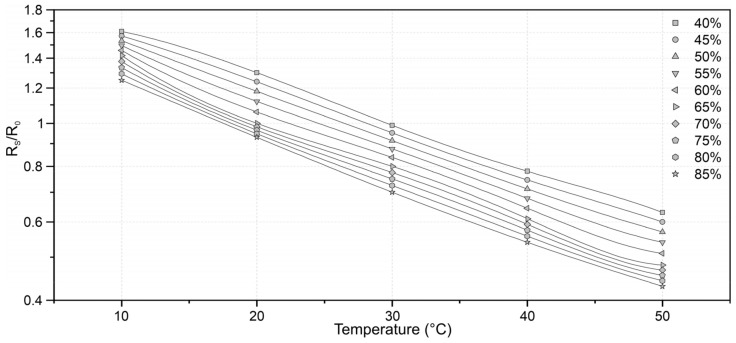
Dependence curves of R_s_/R_0_ in relation to the temperature and relative humidity of the air used for signal correction.

**Figure 6 sensors-21-07456-f006:**
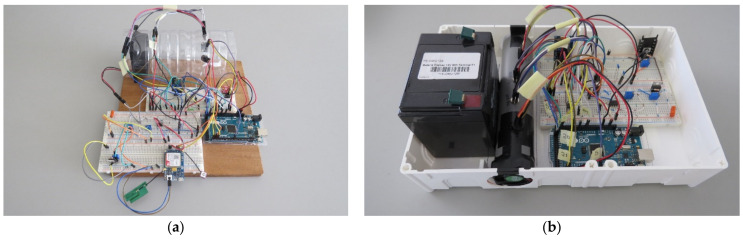
(**a**) System using breadboards with a fixed power supply; (**b**) portable system using a battery.

**Figure 7 sensors-21-07456-f007:**
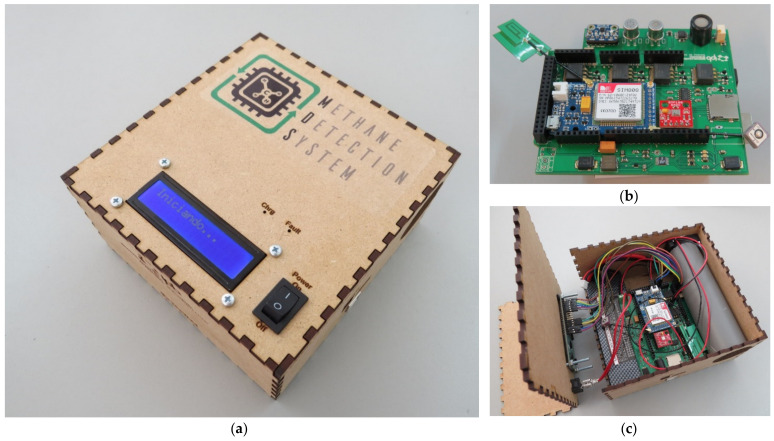
(**a**) Final system inside the wood casing with an LCD display and power button at the top; (**b**) PCB with all components and sensors fixed and welded, the PCB was mounted on top of the Arduino with stackable pins; and (**c**) Distribution of the various elements inside the casing, on the left side the battery, in the middle the PCB and on the right side the air transport system.

**Figure 8 sensors-21-07456-f008:**
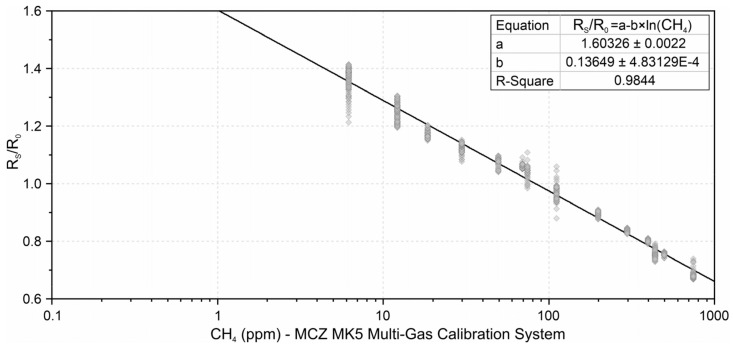
TGS2600 calibration curve using the MCZ-MK5 multi-gas calibration system with different span concentrations.

**Figure 9 sensors-21-07456-f009:**
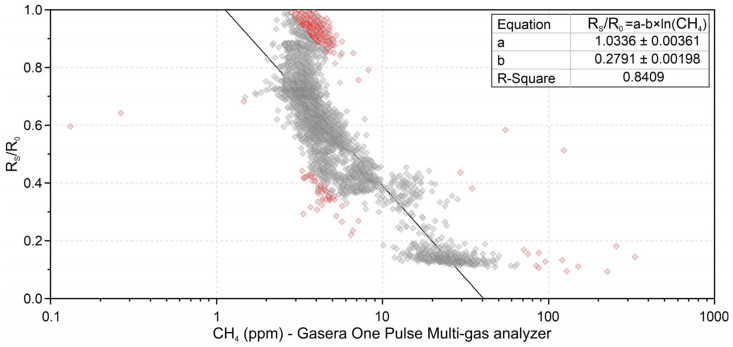
TGS2600 intercomparison curve with a Gasera One Pulse gas analyzer. The points marked in red correspond to the identified outliers, which were not considered for the construction of the curve.

**Figure 10 sensors-21-07456-f010:**
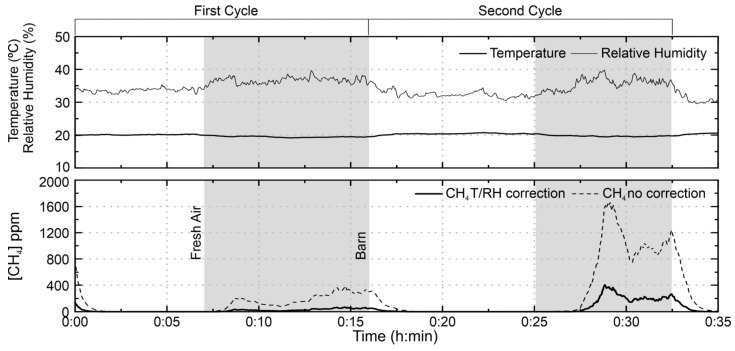
TGS2600 test on 20 April 2017 in the small ruminant barn of the Higher School of Agriculture of the Polytechnic Institute of Bragança. The test was performed in two cycles, each integrating measurements outdoors and inside the barn without animals. The shaded area corresponds to the sensor inside the barn, while the white area corresponds to the outdoor atmosphere (fresh air).

**Figure 11 sensors-21-07456-f011:**
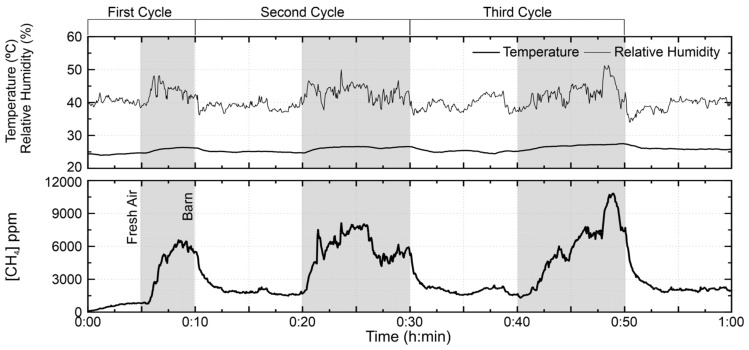
TGS2600 test on 21 April 2017 in the small ruminant barn of the Higher School of Agriculture of the Polytechnic Institute of Bragança. The test performed in three cycles, each integrating measurements outdoors and inside the barn with (quiet) animals. The shaded area corresponds to the sensor inside the barn, while the white area corresponds to the outdoor atmosphere (fresh air).

**Figure 12 sensors-21-07456-f012:**
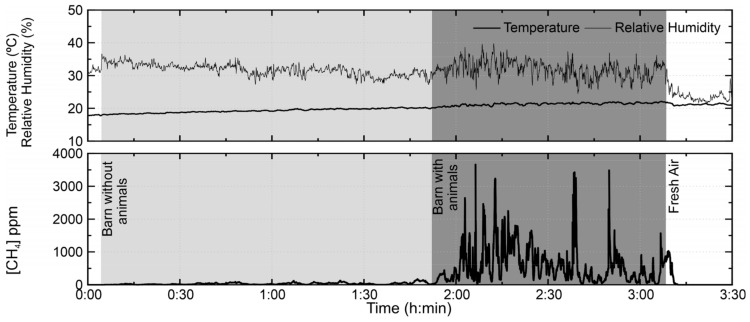
TGS2600 test on 15 May 2017 in the small ruminant barn of the Higher School of Agriculture of the Polytechnic Institute of Bragança. The test was performed covering three situations: the monitoring outside the barn (white area), inside the barn without animals (light grey area) and inside the barn with moving and restless animals (dark grey area).

**Figure 13 sensors-21-07456-f013:**
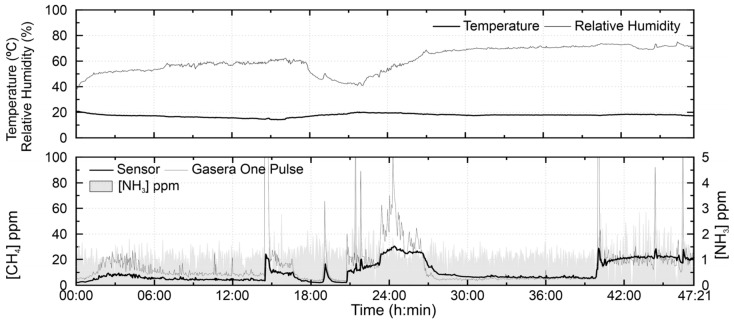
TGS2600 test on 18 October 2020, in a wastewater sludge dewatering room of a wastewater treatment plant.

**Figure 14 sensors-21-07456-f014:**
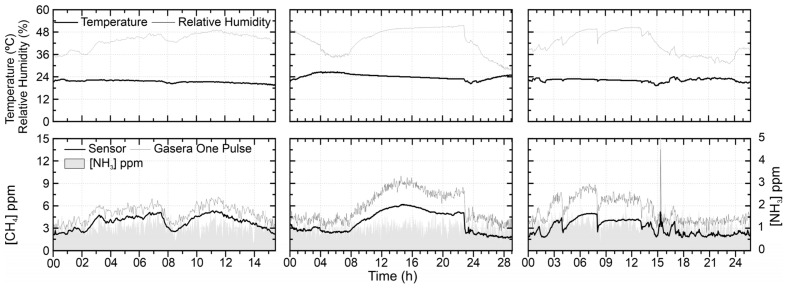
TGS2600 test on 2, 10 and 13 October 2020 performed indoors at the laboratory and office.

## Data Availability

The data that support the findings of this study are available from the corresponding author upon reasonable request.

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
