# Peer review of "A Portable Device for Methane Measurement Using a Low-Cost Semiconductor Sensor: Development, Calibration and Environmental Applications"

_sensors, 2021, doi:10.3390/s21227456_

Round 1
Reviewer 1 Report
The author has developed a prototype of a portable system for methane measurement. Under different application scenarios, the system can complete basic functions. The system adopted the current mature technology, completed the construction of the system, and carried on the experiment with the system. However, the manuscript lacks some theory, especially in the calibration part does not highlight the superiority of the method. The author needs further modification in the following aspects:
(1)Related research is a common problem in the field of gas detection. What is the contribution of the research involved in this manuscript?
(2)What is the calibration method adopted by the author? The description is too succinct and unreadable. In particular, Figures 8 and 9.
(3)What is the exact concentration of methane in the experiment? Why not compare the measured value with the real value to highlight the good performance of the system?
(4)Detection of methane at low concentration is a practical problem and an important research direction of greenhouse gas detection. It is suggested that the author can conduct experimental research on detection of methane at low concentration. In addition, there will be some interference gas in the actual working environment, so the author is requested to eliminate the interference as far as possible to prove the effectiveness of the system. Otherwise, the author needs to study the detection method of mixed gas.
(5)In addition, some details need the author's attention. For example, there is a clerical error in Fig.10 in line 318.
Reviewer 2 Report
This manuscript contains technical details of the development of a portable, low-cost methane gas measurement system that can be used in livestock barns where there are sources of carbon monoxide, isobutane, ethanol, and hydrogen. The contents are aimed at meeting the requirements of the field, such as the effects of relative humidity and temperature on the sensor response and the proposal of a signal correction method considering cross sensitivity. In addition to the construction of a specific gas introduction route to introduce the air in the barn to the semiconductor sensor, the paper describes technologies that will serve as guidelines for future gas monitoring systems, such as the acquisition of sensor response signals via IoT devices. The content of this paper is advantageous in that semiconductor gas sensors can easily realize low cost and low power consumption. Therefore, we conclude that this manuscript is worthy of publication in the Journal of Sensors. However, the authors should answer the following questions before publication.
(1) The authors should indicate the target specifications of the sensors required for livestock barns.
(2) The degree of achievement of this research should be shown for the above 1).
(3) Ammonia is generated as a gas in barns when manure is used. Ammonia is a reducing gas, and methane is also a reducing gas. Of the two, it is impossible to say that only methane is detected.
(4) Fig. 3: The SHT31 should be set upstream of the gas flow and the TGS2600 should be set downstream of the gas flow, but why was it installed in the opposite direction?
(5) It is stated that the error is within 50%, but this is too large. A specific method for calculating the error should be provided as supplementary material. In the case of gas monitoring in barns, the definition of the error should be reconsidered because the situation is always changing.
Reviewer 3 Report
In this manuscript the authors describe a simple and low cost methane sensing platform, based on commercial sensors, that works in different environmental conditions. The effort of the atuthors to create a low cost platform and to take into account temperature and humidity is relevant, although all the components are commercial and the novelty is really low.
I would appreciate an answer from the authors to the following questions/suggestions:
1) Line15: why "in this way"? The abstract must be rewritten, it's not clear, it seems more like a list of things, the sensor is introduced but it's not explained what kind of sensor is.
2) Line 66: What kind of semiconductor? Probably it is better to say "sensors based on semiconductors".
3)Line 98: "It was decided to use the TGS2600 sensor from Figaro based on the global background methane concentration that is around 1.8 ppm" , the correlation between the choice and the reason it's not clear in the text.
4)The nature of the semiconductor should be made explicit in the text.
5)Figure 9 have to be better explained in the text. The fit is questionable.
6) How are the results obtained in the barn comparable with other devices, like Gasera, especially in terms of stability?
7) The conclusions seems to suggest that this work is very preliminary.
A general comment: the quality of the figures must be improved and the numbers in the plots must be bigger.
Round 2
Reviewer 1 Report
This version can be published.
Reviewer 3 Report
The authors answered all the questions.